# Choosing fast and simply: Construction of preferences by starlings through parallel option valuation

**Tiago Monteiro** [1] *, **Marco Vasconcelos** [2] *, **Alex Kacelnik** [1] *

**1** Department of Zoology, University of Oxford, Oxford, United Kingdom, **2** William James Center for Research, University of Aveiro, Aveiro, Portugal

* tiago.monteiro@zoo.ox.ac.uk (TM); mvasconcelos@ua.pt (MV); alex.kacelnik@zoo.ox.ac.uk (AK)

## Abstract

The integration of normative and descriptive analyses of decision processes in humans struggles with the fact that measuring preferences by different procedures yields different rankings and that humans appear irrationally impulsive (namely, show maladaptive preference for immediacy). Failure of procedure invariance has led to the widespread hypothesis that preferences are constructed "on the spot" by cognitive evaluations performed at choice time, implying that choices should take extra time in order to perform the necessary comparisons. We examine this issue in experiments with starlings (*Sturnus vulgaris*) and show that integrating normative and descriptive arguments is possible and may help reinterpreting human decision results. Our main findings are that (1) ranking alternatives through direct rating (response time) accurately predicts preference in choice, overcoming failures of procedure invariance; (2) preference is not constructed at choice time nor does it involve extra time (we show that the opposite is true); and (3) starlings' choices are not irrationally impulsive but are instead directly interpretable in terms of profitability ranking. Like all nonhuman research, our protocols examine decisions by experience rather than by description, and hence support the conjecture that irrationalities that prevail in research with humans may not be observed in decisions by experience protocols.

## Introduction

The distinction between 2 kinds of processes underlying human choice [1]—a fast one (System 1), presumably shared with other species, and a slower one (System 2), involving cognitive evaluation of alternatives at the time of choice—is widely accepted [2–6] but has not yet been integrated with decision processes in other animals.

Something similar occurs with the failures of procedure invariance detected when human preferences are measured by direct rating (*viz* "willingness to pay") or by choice [7], with measures of impulsiveness [8–11] (defined as a stronger preference for smaller–sooner over larger–later rewards than explained by an optimality criterion such as rate of gains over time) and with the distinction between "decisions from description" and "decisions from

**Data Availability Statement:** All data can be found in the S1 Data file.

**Funding:** This research was supported by Grant BB/G007144/1 from the UK Biotechnology and Biological Sciences Research Council to AK. TM

was supported by a doctoral grant from the Portuguese Foundation for Science and Technology (SFRH/BD/43385/2008) and a Pembroke College Graduate Scholarship. MV was supported by the Portuguese Foundation for Science and Technology (UIDB/04810/2020). The funders had no role in study design, data collection and analysis, decision to publish, or preparation of the manuscript.

**Competing interests:** The authors have declared that no competing interests exist.

**Abbreviations:** DDDM, drift-diffusion decision model; SCM, sequential choice model.

experience" [12, 13]. Failures of procedure invariance are critical, because they are used to support the widespread notion that preference does not exist outside choice contexts but is constructed "on the spot" through comparative evaluations at choice time [7].

We believe that all these issues are deeply interconnected and that investigating decision-making in other species with reference to hypotheses developed for humans, and vice versa, should help to develop an integrated and more general theory of decision-making.

We contribute to this integration and present critical experiments with starlings (*S. vulgaris*) as research species. Our main experimental findings are that (1) ranking of a measure of direct rating (defined as any graded response to individual options outside a choice context; in humans, a common direct rating metric is "willingness to pay," and in starlings, we use response time) accurately predicts preference in choice, overcoming reported failures of procedure invariance and claims that the valuations underlying preferences at the time of choice do not exist outside choice contexts; (2) response times in choices are shorter than when only 1 option is present, contradicting the hypothesis that a cognitive comparative evaluation occurs on the spot at choice times; and (3) our subjects' choices are not irrationally impulsive but instead are directly interpretable in terms of profitability ranking. Like all nonhuman research, our protocols examine decisions by experience rather than by description and hence, support the conjecture that irrationalities that prevail in results of human decision-making (including failures of procedure invariance) may be endemic to the so-called System 2 peculiar to our species. For an in-depth review of the empirical significance of using experience or description in human decision research see [13].

Our results confirm hypothetical models of choice that originate in foraging research. In nature, the probability (per unit of time) of encountering 2 or more independently distributed prey types simultaneously reflects the product of their probabilities, meaning that simultaneous choice must be comparatively rare. This might not be strictly so when the distribution of prey types is not independent (as when preying upon gregarious prey), but even then, the ability to select between multiple profitable prey can build upon efficient hunting of prey found on their own, whereas the reverse is not true. For these reasons, we expect stronger selection to act adaptively in sequential than in simultaneous encounters, and the psychological machinery evolved to act efficiently in sequential encounters to be biologically more significant than specific adaptations for simultaneous choice. This rationale underlies the sequential choice model [14, 15] (SCM, see S1 Text for mathematical details and numerical simulations). The SCM can be construed as a race (inhibition-free) drift-diffusion decision model (DDDM, [3, 16–20]) but with the special feature of predicting choice as a consequence of mechanisms driving action in sequential encounters. It postulates that when facing a response opportunity, the subject undergoes an accumulation process leading to action when a threshold is crossed, with the rate of accumulation and the threshold determining the distribution of response times for each prey type, depending on its profitability relative to the background rate of consumption. A difference with other DDDMs is that in the SCM the hypothetical accumulation is not interpreted as a gathering of evidence in favor of one option over the alternative but a build-up leading to action toward a stimulus regardless of whether it is alone or in the presence of alternatives. Following classical optimal foraging theory [21–25], we define profitability of a prey type as the ratio of gain (typically net energy content) to involvement time (time to pursue and handling each item), excluding time costs common to all prey types, such as travel or searching periods. Based on empirical evidence, the interval between encountering a prey type and attacking it (which has no direct interpretation in foraging theory) is modeled as being variable, with distributions' parameters depending on both the prey type's profitability and the rate of gain in the background: relatively richer prey are attacked faster. Critically, the SCM postulates that when more than 1 potential prey is present, a "choice" results from the

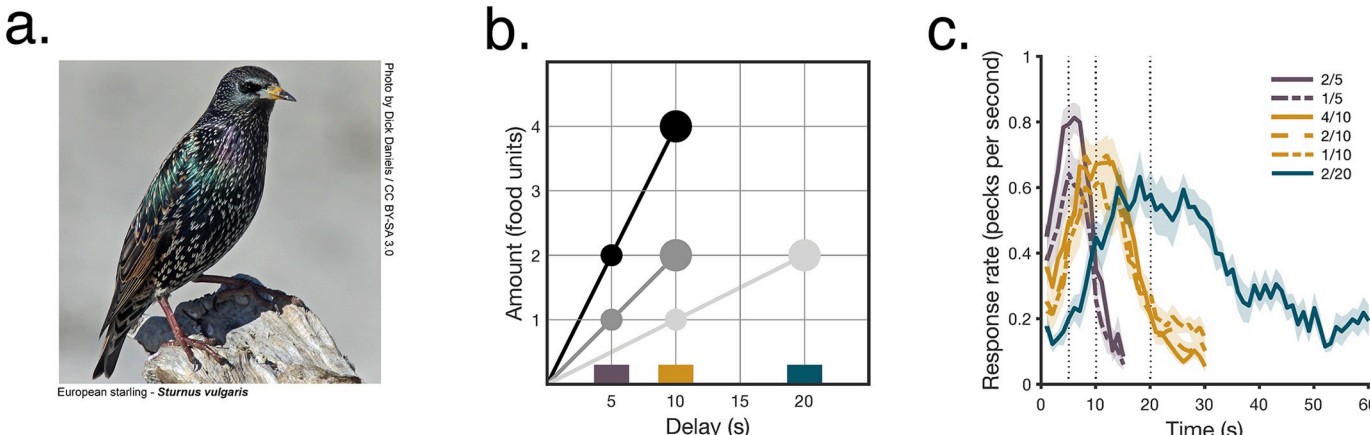

**Fig 1. Starlings were trained in a multialternative environment. a**. European starling (*S. vulgaris*, [*photo credit*: Dick Daniels, http://carolinabirds.org/]/CC *BY-SA* https://creativecommons.org/licenses/by-sa/3.0/]). **b**. Alternatives present in the experimental environment. Each symbol depicts a profitability ratio (a specific combination of amount and delay) and each shading, a profitability (a given ratio of amount to delay in rewards/second). Lines join stimuli with equal profitability. Within each profitability pair, the larger symbol indicates the Larger–Later profitability ratio and the smaller symbol the Smaller–Sooner one. **c**. Rate of responding in probe trials (mean ± SEM, *n* = 9 birds). All data included in this figure can be found in the S1 Data file.

independent process for each of the options present, without any "on the spot" comparison, without building evidence about relative values as assumed by classical DDDMs, and without construction of preference at the time of choice.

A consequence of these assumptions is that response times toward a given stimulus when picked out of a choice should be shorter than when the same stimulus is encountered alone (i.e., if prey types A and B are encountered simultaneously, and A is attacked, observed response time should be shorter than when A is encountered in the absence of an alternative). Further, the shortening effect should be stronger for leaner, infrequently chosen stimuli. These predictions result from assuming cross-censorship between independent processes for each of the options present and contradict the hypothesis that preference results from a comparative evaluation at choice time and hence, involves extra time. The mathematical justification for these properties is detailed in S1 Text.

We trained starlings (Fig 1a) in an environment in which 6 stimuli, each associated with a given profitability ratio (Fig 1b), were randomly encountered, either sequentially or in binary pairings requiring a choice. Notice that we make a nomenclature distinction between "profitability ratio" and "profitability." We define profitability ratio as particular combinations of an amount and delay to food (thus, 1 pellet/10 seconds and 2 pellets/20 seconds are 2 different profitability ratios), and profitability as the quotient between them (0.1 pellets/second in both cases; this is the variable used in classical foraging theory [25]). This distinction is necessary because profitability is scale free, but options with the same profitability (as in symbols joined by lines in Fig 1b) can impact preferences if they differ in profitability ratio, and the distinction is useful to discuss impulsivity. We used 6 profitability ratios that allow the following comparisons: 3 had equal delay but differed in amount, 3 had equal amount but differed in delay, and 3 pairs had equal profitability but differed in both amount and delay. In addition to preference in binary choices, we recorded response times in single-option encounters and in the 30 types of binary-choice outcomes (6 options allow for 15 possible pairings, and in each pairing, the subject could choose either of the 2 options). We also interspersed infrequent unrewarded probe trials to assess the animals' sensitivity to the delay associated with each stimulus. These probe trials (Fig 1c, key in colored rectangles in Fig 1b) showed that the birds learned these delays and that their timing was not distorted by differences in amount.

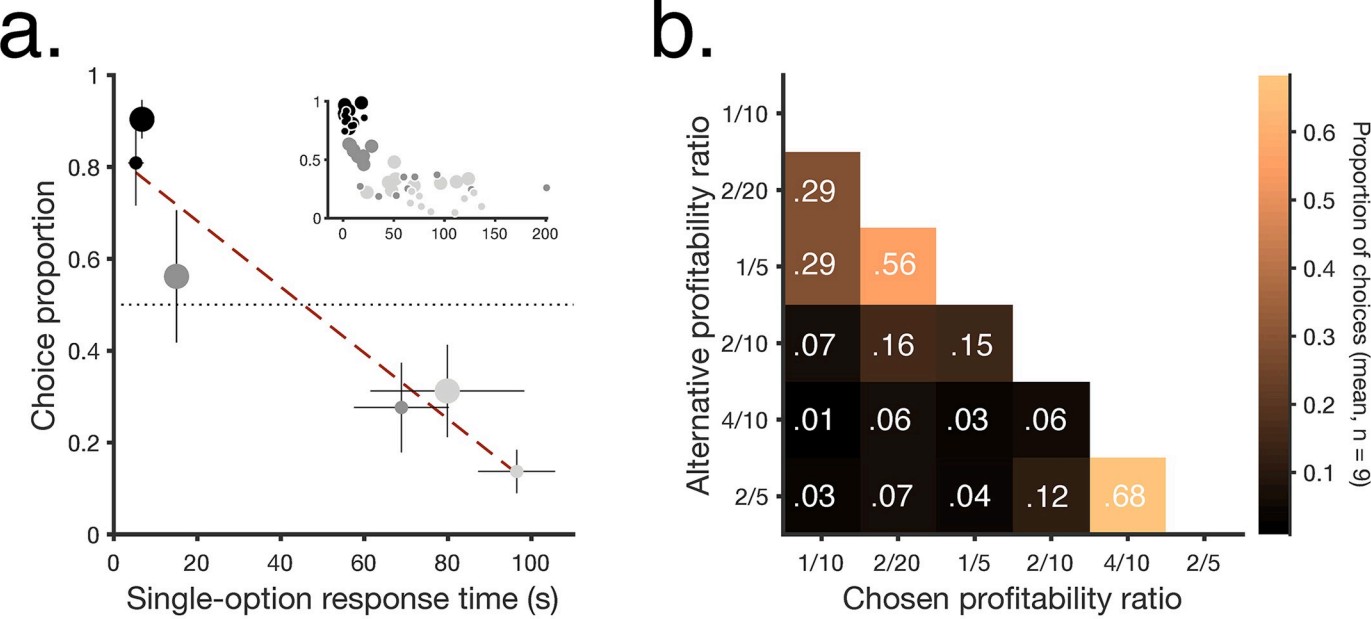

**Fig 2. Profitability impacts response times and choices. a**. Preference in choice trials (mean proportion picked when presented paired with any of the other 5 profitability ratios) versus response time in single-option trials (means ± SEM, *n* = 9 birds). Inset shows individual data. Symbols match panel in Fig 1b. The dashed line shows a Deming regression (Estimated model: *y* = −0.0072*x* + 0.8259, [slope = 95% CI −0.004 to −0.0104]). **b**. Mean proportion of choices in every binary-choice combination (*n* = 9 birds). All data included in this figure can be found in the S1 Data file.

As assumed by the SCM, higher profitability was associated with shorter response times in single-option trials and with higher preference in choices (Fig 2a). Moreover, the strength of preference for each option was modulated by the profitability of the alternative option (Fig 2b). A unique, specific prediction of the SCM is that response times in choices should be shorter than in single-option trials, with the effect being stronger for leaner alternatives (see S1 Text). To test for this, Fig 3a shows the difference in response time for each stimulus type when picked out of a pair compared with the same stimulus type in single-option trials. Average response times were shorter or equal in choices than in single-option encounters (Fig 3a, bottom row; Fig 3b shows further quantitative details), as predicted by the SCM, and the effect was stronger for the least-preferred alternatives. The figure also shows that this population result reflects the data from individual subjects. A bootstrapped analysis confirmed that these results, even at the individual level, were not due to chance (Fig 3c and 3d, see Analyses section for details).

The common assumption that decision-makers show an impulsive bias to prefer smaller–sooner rewards over equally profitable larger–later ones is explored in Fig 4a. In the pairwise comparisons in which the profitability of the options was equated, the birds, in fact, showed the opposite: They preferred the larger–later alternative over the smaller–sooner one. This result contradicts prevalent ideas on impulsivity [9] but is consistent with reward rate maximization with partial account of the common time intervals [26–29] (Fig 4b).

In summary, we show procedure invariance between direct rating and choice when using response time as a direct rating metric, and we find that the SCM is appropriate to describe the choice process. This contrasts with the failure of procedure invariance characteristic of human experiments, but this does not necessarily imply species' differences. It is worth recalling that the discovery of failures of procedure invariance in humans are typical of experiments based on description rather than experience, and such experiments are not possible in nonhumans.

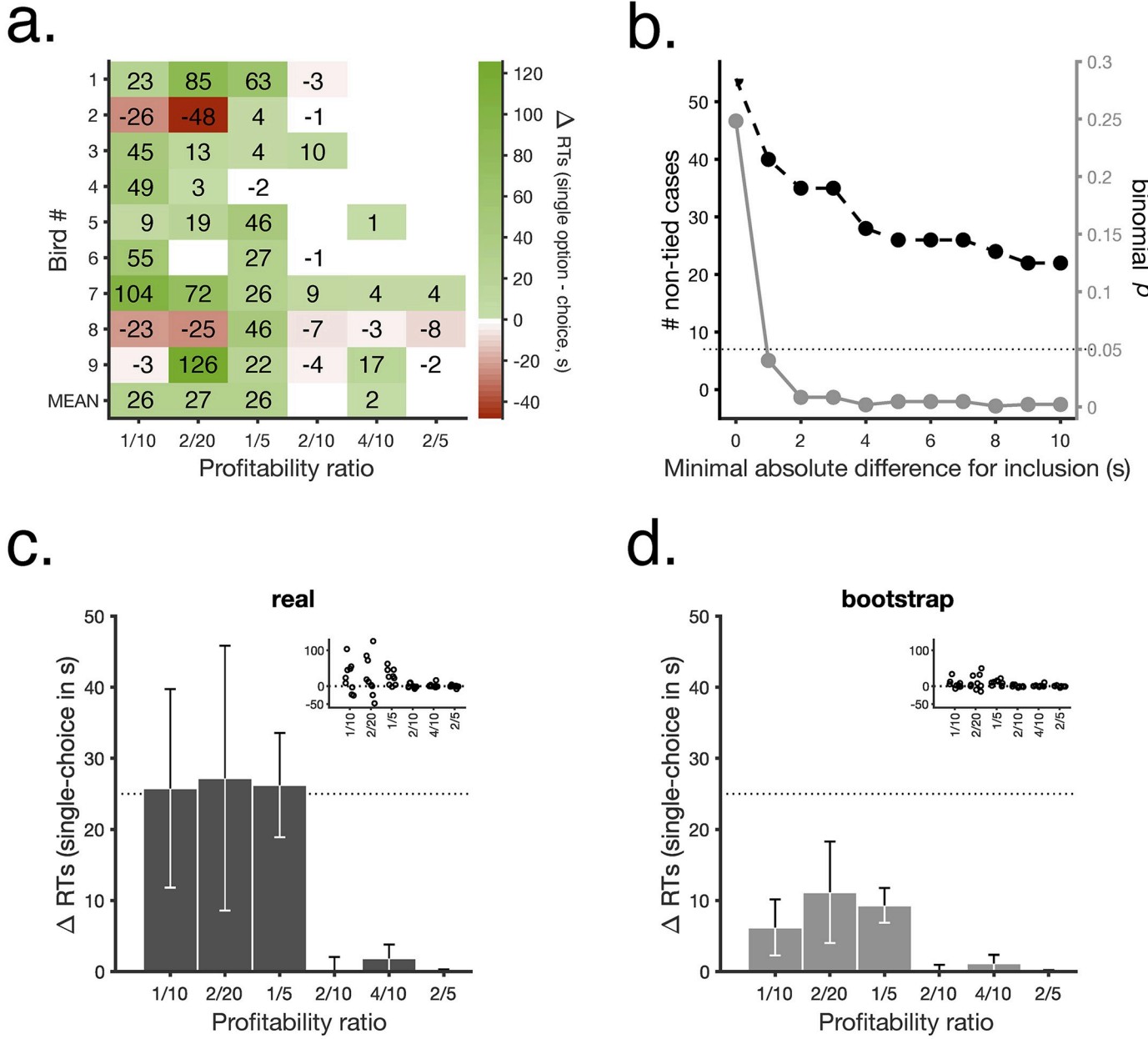

**Fig 3. RTs are shorter in choices. a**. ΔRT toward each profitability ratio between single-option and choice trials, averaged against the other 5 combinations; bottom row shows means across subjects. White cells are tied cases in which differences were lower than 1s in either direction. **b**. Number of nontied cases (out of 54) as a function of increasing minimal absolute response time difference (black) and corresponding binomial $p$-value (gray). **c**. Mean differences of individual mean response times in single-option trials minus individual mean response times in choice trials ($n = 9$, mean ± SEM). Inset shows individual animal means, with symbols laterally displaced for visualization purposes. (same data as bottom row in **a**.) **d**. Same as in **c**. for bootstrapped differences. A repeated-measures ANOVA, with option (6 levels) and difference as factors (2 levels), and mean ΔRTs as dependent variable, showed a significant effect of difference, only (Greenhouse–Geisser adjusted $F(1,8) = 7.307$, $p = 0.027$, $\eta^2_P = 0.660$). All data included in this figure can be found in the S1 Data file. RT, response time; ΔRT, difference in response time.

It is thus possible that such failures are peculiar of human protocols using description rather than experience. Whether this or genuine species differences in decision mechanisms are responsible for the contrast between our results and those reported in humans remains to be determined. These possibilities can be differentiated by replicating our procedures with

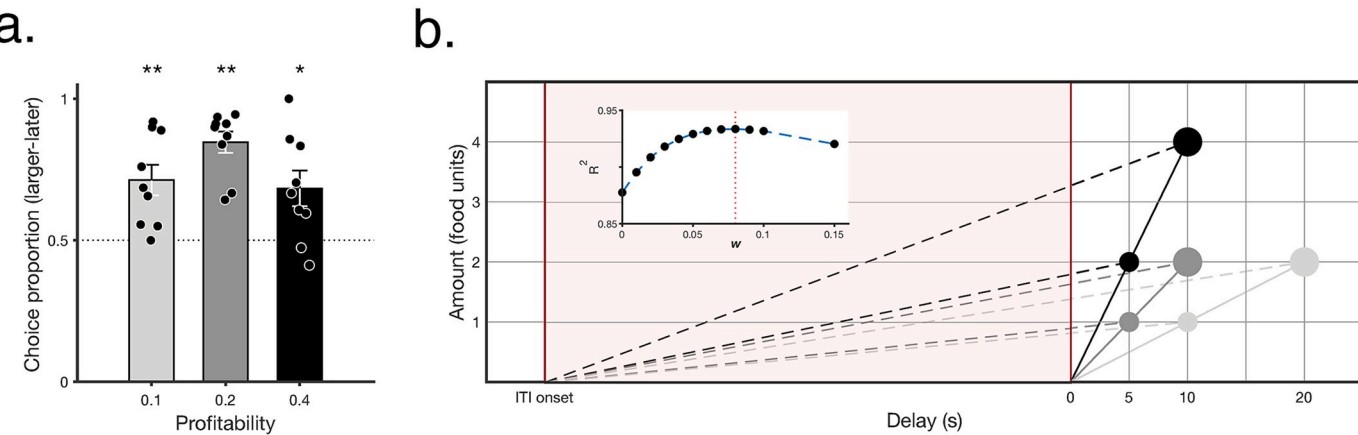

**Fig 4. When profitability is equated, starlings prefer larger–later alternatives. a**. Proportion larger–later alternative chosen against the equal profitability smaller–sooner combination (mean ± SEM, $n$ = 9 birds). Dots show individual animals, laterally displaced for visualization purposes. Two-tailed $t$-tests against 50%, smallest $t(8)$ = 2.912, $^*p < 0.05$, $^{**}p < 0.01$, smallest Cohen's d = 2.05, effect size = [0.71, 0.96]. **b**. Long-term rate of returns R in which the time in the ITI is given the same decision weight as the delay between response and outcome (R = amount/(ITI + delay); equivalent to the slope of the dashed lines) and profitability as in Fig 1b (slope of solid lines). Inset shows $R^2$ of linear regressions between observed preference ranking (mean across animals, $n$ = 9) and long-term estimated rate of returns when the common ITI is given a discounted weight (R = amount / ($w$*ITI + delay), computed with increasing $w$); note that the strongest predictive power is for a long-term rate computation that includes an 8% weight of the ITI. All data included in this figure can be found in the S1 Data file. ITI, intertrial-interval.

human subjects. If humans behave in such protocols as starlings do, the problem becomes mapping the significance of the finding in human experience. It is possible that the processes we report, and the foraging rationale that explains them, are relevant in humans, but only for problems recruiting the so-called System 1 of human decision-making and that they are consistent with the evidence for different neural representation of the values of alternatives and the background [30, 31] but are not relevant to the reasoning-based decision process that prevails when System 2 is in action.

## Materials and methods

### Ethics statement

All experiments were approved by the Department of Zoology Ethical Committee, University of Oxford, and were carried out in accordance with the current laws of the United Kingdom. Animals were obtained under English Nature license No. 20010082 and were cared in accordance with the University of Oxford's "gold standard" animal care guidelines. Maintenance and experimental protocols adhered to the Guidelines for the Use of Animals in Research from the Association for the Study of Animal Behaviour/Animal Behavior Society [32].

On completion, the birds were reintroduced into communal aviaries and eventually released into the wild.

### Subjects

Twelve adult wild-caught European starlings (*S. vulgaris*) participated in the current experiment. All subjects were naive to experimentation. Prior to testing, birds were housed communally in a large outdoor aviary (3 meters high × 3.2 meters wide × 4 meters long) where they received ad libitum food (a mixture of turkey crumbs and mealworms (*Tenebrio sp.*) and had continuous access to fresh drinking and bathing water.

During the course of the experiment, starlings were moved in pairs to indoor cages, which were visually but not acoustically isolated. Indoor temperatures ranged from 16 ˚C to 18 ˚C,

and lights followed a 12:12 dark schedule with daylight at 07h00, and gradual transitions at dawn and dusk. Starlings were rewarded with precision pellets throughout the experimental session and provided with 4 hours (13h00–17h00) of free access to a mixture of turkey crumbs and Orlux Remiline universal granules, mealworms, and social interaction with the respective pair. This regime maintains starlings at approximately 90% of their free feeding weight. Birds had continuous access to drinking water and received bathing trays every day.

## Apparatus

Each indoor cage (1,350 mm × 784 mm× 800 mm [l × w × h]) was composed of 2 units, vertically mounted (800 mm each). Each unit included 2 experimental areas separated by a common middle section. Each experimental area had a panel attached 100 mm above the floor (S1a Fig). The panel was 400-mm tall with 3 sections: a middle subpanel, facing the cage (115-mm wide), and 2 side subpanels (same length) attached to the cage at a 120˚ angle from the center subpanel. The middle subpanel had 1 response key in the center (110 mm from the bottom) and the food hopper (25 mm from the bottom), which was connected to the pellet dispenser (Campden Instruments) containing 20-mg precision dustless pellets. Each side subpanel had 1 response key in the center (110 mm from the bottom). Every key was composed of a 16-LED lights matrix that could display 16 different symbols in 7 possible colors (S1b Fig). Trials were governed by custom software running on the Microsoft Windows operating system and attached to an Animal Behaviour Environment Test System (Campden Instruments) via WhiskerServer [33].

## Pretraining

Following magazine training, in the first phase of pretraining, the birds were trained to peck a flashing white center key (700 milliseconds ON, 300 milliseconds OFF). A single peck to this key delivered 4 precision pellets, extinguished the keylight, and initiated a 120-second inter-trial-interval (ITI). Failing to peck the key within 30 seconds from the onset of flashing resulted in the same events as a peck. Training thus combined autoshaping with an operant response. Once birds were reliably pecking the center key, the lateral keys were introduced. Trials again began with the flashing of a white center key. A peck to this key turned its light off and started flashing a white lateral key. A peck to the flashing lateral key turned it off and produced 2 food pellets followed by a 90-second ITI. During this phase, failing to peck the center key within 30 seconds from trial onset resulted in a 10-second period with the light switched off, followed by the ITI. Failing to peck the flashing side key during the same time period was followed by food 75% of the time. This program was maintained 5 hours per day until the birds reliably pecked the center and side keys on at least 80% of the trials.

Next, birds received response-initiated fixed interval (FI) training on the side keys. In this phase, 2 pecks to the side key were required to deliver food. The first peck turned the symbol steadily on and began a delay, the duration of which was progressively increased from 1 to 20 seconds. The first peck after the delay elapsed extinguished the keylight, delivered 2 food pellets, and initiated a 45-second ITI. Once birds were reliably obtaining food on FI 20-second trials, the experimental sessions began.

For experimental sessions, a different set of symbols/colors were used for each bird.

## General procedure

Starlings were trained in an environment consisting of 6 different options composed by a different amount/delay to food combinations (2 pellets in 5 seconds, 4 pellets in 10 seconds, 1 pellet in 5 seconds, 2 pellets in 10 seconds, 1 pellet in 10 seconds, and 2 pellets in 20 seconds);

hence, each option signaled 1 out of 3 possible profitability ratios (0.4, 0.2, and 0.1 pellets per second, Fig 1b).

Each training session lasted a maximum of 5 hours and was composed of 252 trials (240 single-option trials, 40 per option, 20 in each possible location; and 12 probe trials, 2 per option, 1 in each possible location; see details next).

Birds were moved to a testing phase after completing 15 days with more than 75% completion of daily trials or after 20 days of 60% of daily completion, whichever came first. In testing, single-option trials were interspersed with binary-choice trials between all possible combinations of different options. Testing sessions were also 252 trials and/or 5 hours long (120 single-option trials, 20 per option, 10 in each possible location; 120 binary-choice trials, 8 per pairing, 4 in each possible location; and 12 probe trials, 2 per option, 1 in each possible location). Testing ended after 2 blocks of 3 consecutive days with no trend in preferences (and SD < 0.15) or after 15 days, whichever came first.

## Trial types

Single-option trials began with the flashing illumination of the center attention key. A peck to this flashing key turned the key off and began flashing (700 milliseconds ON, 300 milliseconds OFF) 1 of the 2 side keys (randomly assigned). A peck to the illuminated flashing side key turned the key steadily on and initiated its programmed delay. Once this interval elapsed, the first peck to the key turned the keylight off and delivered the predefined number of pellets of reward at a rate of 200 milliseconds per pellet, after which a 45-second ITI was initiated. Not pecking the keys at any stage (flashing or steady keys) meant not moving forward to the next trial.

Probe trials were as single-option trials, but once pecked, a given symbol remained on for 3 times its typical delay. At the conclusion of this interval, the light extinguished without reward or requiring a peck, followed by the ITI [34, 35].

Choice trials offered individuals simultaneous choices between combinations of pairs of training stimuli, chosen among 15 possible binary-combinations of the 6 alternatives. Trials again began with a flashing center attention key. A peck to this key turned it off and began flashing the 2 side keys (each one displaying a different symbol). The location of presentation was also randomized. A peck to one of the flashing keys turned the selected key on and initiated its programmed contingencies and turned the unselected key off. Thus, the bird committed itself to a given option with a single peck. Absence of pecking produced the same consequences as in single-option trials.

## Data selection

Only the last 6 testing days (i.e., for preferences and response times) were considered in the analysis. Three birds were excluded from all analyses after repeatedly failing to reach the experimental criteria (see General procedure).

## Analyses

All analyses were run in custom MATLAB R2019a software (Mathworks), available upon request.

Deming regression analysis in Fig 1d uses code from [36], retrieved from MATLAB Central File Exchange. (https://www.mathworks.com/matlabcentral/fileexchange/33484-linear-deming-regression).

We are interested in the difference in response times for each option when picked up of a choice as opposed to when the option is the only one available. The "construction of

preference" hypothesis assumes that choices should take time, so reaction times should be longer, whereas the SCM predicts that reaction times in choices should be shorter. As Fig 3a shows, reaction times were predominantly shorter in choices, and the present analysis aims at testing whether the observed results could be outcome of chance. To this effect, we performed a bootstrap analysis (Fig 3c and 3d) in which we considered the null hypothesis that both single-option and choice reaction times are sampled from the same distribution, namely, that there is no systematic difference between them. Under this null hypothesis, for each animal and option, all response times collected during this experiment (i.e., from single-option and choice trials) come from the same underlying distribution. If that is the case, any difference between average response times in single-option versus choice trials (i.e., ΔRT) is simply due to chance.

To test the null hypothesis, we generated subject-specific null distributions of response times. For each animal and option, we pooled response times for single-option and choice trials, so that each animal had 6 pools of response times, one per option. Then, we simulated $10^5$ experiments per option, where in each iteration, we generated $n$ choice response times (where $n$ was the animal's real number of choices for that option), drawn with replacement from the response time pool corresponding to that option. Next, we calculated the average for each of these simulated distributions and subtracted these averages from the corresponding real (observed) mean single-option response time (i.e., bootstrapped difference).

For example, bird number 1 gave us 77 response time values for option 2/20 in single-option trials and 46 response times from its choices of option 2/20 in choice trials. So we formed a single pool of 77+46 response times for that subject with its real response times. In each simulated iteration, we randomly selected 46 response time values from its pool, computed the average ($n$ = number of simulations) and subtracted this mean from the mean of observed reaction times in that subject's choices of option 2/20 − ΔRT.

Finally, we compared the simulated ΔRT with the observed ΔRT using a repeated-measures ANOVA.

## Supporting information

**S1 Text. The SCM.** Computational description, analytical implementation and numerical simulations. SCM, sequential choice model.
(PDF)

**S1 Fig. a**. Schematics of the operant panels (adapted with permission from [15], APA). **b**. Close-up photographs of the operant panels displaying different symbols (top) and colors (bottom).
(TIF)

**S1 Data. An Excel file with all data used in the main figures, with each tab corresponding to a figure panel, from main Figs 1 to 4, respectively.** Tabs from the same figure are labelled with the same color.
(XLSX)

## Acknowledgments

We thank Simon and Jan Parry for trapping our participants. We would like to thank Miguel Santos, Andreia Madalena, Filipe Rodrigues, Catarina Albergaria, Bruno Cruz, Sofia Soares, Andrés Ojeda, Mark Walton, and Alasdair Houston for feedback on earlier versions of the manuscript.

## Author Contributions

**Conceptualization:** Tiago Monteiro, Marco Vasconcelos, Alex Kacelnik.

**Data curation:** Tiago Monteiro.

**Formal analysis:** Tiago Monteiro, Marco Vasconcelos.

**Funding acquisition:** Tiago Monteiro, Marco Vasconcelos, Alex Kacelnik.

**Investigation:** Tiago Monteiro, Marco Vasconcelos, Alex Kacelnik.

**Methodology:** Tiago Monteiro, Marco Vasconcelos, Alex Kacelnik.

**Project administration:** Tiago Monteiro.

**Resources:** Alex Kacelnik.

**Software:** Tiago Monteiro.

**Supervision:** Marco Vasconcelos, Alex Kacelnik.

**Writing – original draft:** Tiago Monteiro.

**Writing – review & editing:** Tiago Monteiro, Marco Vasconcelos, Alex Kacelnik.

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
