## [Editor Report · Decision Letter 0]

3 Feb 2020

Dear Dr Monteiro, 

Thank you for submitting your manuscript entitled "Thinking fast and simple: how decisions are taken" for consideration as a Short Report by PLOS Biology. Please accept my apologies for the delay in sending this decision to you. We were interested in your study, and thus, sought advice from an Academic Editor with relevant expertise. With that advice now in hand, I am writing to let you know that we would like to send your submission out for external peer review.

However, please note that we would need to be persuaded by the reviewers that the findings are as novel and that they extend to human decision-making as directly as you have suggested for further consideration.

In addition, before we can send your manuscript to reviewers, we need you to complete your submission by providing the metadata that is required for full assessment. To this end, please login to Editorial Manager where you will find the paper in the 'Submissions Needing Revisions' folder on your homepage. Please click 'Revise Submission' from the Action Links and complete all additional questions in the submission questionnaire.

Please re-submit your manuscript within two working days, i.e. by Feb 05 2020 11:59PM.

Kind regards,

Gabriel Gasque, Ph.D.,

Senior Editor

PLOS Biology

---

## [Decision Letter · Decision Letter 1]

19 Mar 2020

Dear Dr Monteiro,

Thank you very much for submitting your manuscript "Thinking fast and simple: how decisions are taken" for consideration as a Short Report at PLOS Biology. Your manuscript has been evaluated by the PLOS Biology editors, by an Academic Editor with relevant expertise, and by four independent reviewers. Please accept my apologies for the delay in communicating the decision below to you.

In light of the reviews (below), we will not be able to accept the current version of the manuscript, but we would welcome re-submission of a much-revised version that takes into account the reviewers' comments. We cannot make any decision about publication until we have seen the revised manuscript and your response to the reviewers' comments. Your revised manuscript is also likely to be sent for further evaluation by the reviewers.

We expect to receive your revised manuscript within 2 months. 

**IMPORTANT - SUBMITTING YOUR REVISION**

Your revisions should address the specific points made by each reviewer. As a Short Report, please keep the number of main figures down to four. Please also submit the following files along with your revised manuscript:

*Re-submission Checklist*

*Published Peer Review*

*PLOS Data Policy*

*Blot and Gel Data Policy*

Sincerely,

Gabriel Gasque, Ph.D., 

Senior Editor

PLOS Biology

REVIEWS:

Reviewer #1: Monteiro et all provide a very interesting set of results regarding the nature and mechanisms for decision making by using birds as an animal model. They find that choices are not made on the spot but rather reflect some intrinsic preferences. Additionally, they find that decision making in their task does not involve direct comparison between alternatives, and rather the behavior can be explained by a model where options are selected randomly one from each other. In general, the paper is well written, although I think that the claims about generality to other conditions could be toned down slightly for a more balanced manuscript. 

The description of the SC Model is difficult to follow. It is described in previous literature, but I think that a simplified mathematical description of it in the SI will be valuable for the readers. In addition, one could use that opportunity to develop the basic predictions and their underlying intuitions. 

Indeed, the prediction that RTs should be shorter for pair vs single decisions is a very strong one. However, I have a number of concerns related to its experimental testing. First, the prediction is qualitative, rather than quantitative. However, the SC model should predict by how much one expects to reduce RT. I think that making a semi-quantitative analysis of the model and the data would be important. Otherwise, it is unclear whether short RT times in pair vs single choices provides really support in favor of the SCM and against the DDM, or whether it provides support for a different interpretation of the results (that I develop below).

I am concerned that the shorter RTs observed in the pair choice condition is due to some attentional effects that do not have anything to do -a priori- with the choice mechanics. In the experimental setup, in pair conditions there is more physical stimulation than in single conditions -there are two choices and the amount of light over the display, if I understand well, is also larger. It can happen that more stimulation can increase attention and therefore reducing RTs. I wonder whether if one can control from arousal/attentional effects by controlling for differences in low level features of the stimulus (such as total brightness of the choices between single and pair conditions), or whether actually the authors have already taken care of this.

I have some comments about Figure 1. I would propose using the same color code in panels b and c. Right now it is quite confusing using colors for one panel, and gray levels for the other. In panel e, it is difficult to see the large differences in RT. I think it would be better to plot RT for single vs RT for pairs, and see that this line has lower slope than one, or something on those lines. 

Minor comments:

-I don't see how reference 6 first into supporting Systems 1 vs Systems 2 hypothesis. Also I would say that this distinction is largely controversial, at least many people think that System 1 is not sloppy and rather follows the rule of automatic Bayesian inference.

-In the main text the definition of profitability is not very transparent. I would define the mathematical terms in the text by using symbols, and just spell out the definition of profitability using an equation.

-In the list references 26-28 for reward maximization in sequential decision making, Drugowitsch et al, Journal of Neuroscience, 2012 could be a missing relevant reference.

Reviewer #2: This paper addresses the interesting topic of whether a non-human animal will show processes similar to the fast and slow decision processes famously reported for humans. As discussed, humans have been found to show failures of procedural invariances in their decision making and have been suggested to make different decisions depending on which system is activated.

In this research, starlings learned six options that differed in amount and delay to food. Latency to peck on single option trials was predictive of preference on binary choice trials -showing procedural invariance - and latencies were shorter on choice trials than when a single option was presented - inconsistent with a deliberation process on choice trials. The birds also did not show an irrational tendency to choose smaller sooner rewards. 

The topic is important, the results are clear, and there are interesting messages in the paper, for example that simultaneous decisions are likely rare and less important in nature. However, I have several problems with the current manuscript, some of which likely stem from the short report format that does not provide space to clarify and qualify statements. 

For example, "irrational impulsivity" is introduced in the abstract as a human problem. The end of the abstract suggests that irrationality in humans could be due to the use of description. The authors may be just referring to procedural invariance here but since impulsivity is labelled as irrational this is certainly not clear. I think a reader who doesn't know the literature would conclude from reading the paper that impulsivity is a human problem and that these results suggest that since starlings don't show it in an experience-based tasks that it may be because of the use of described problems for humans. What is not discussed is that other animals also show impulsivity and humans do too in experience-based tasks (e.g. Jimura et al., 2009). So I think more needs to be said about how and why the starlings appear to differ from not only humans on described choices but also other animals and humans on experience-based tasks. 

I am also not clear on what aspect of these results are novel. The main findings stated in the abstract are: "(1) ranking alternatives through direct rating (response time) accurately predicts preference in choice, overcoming failures of procedural invariance; (2) preference is not constructed at choice time nor does it involve time (we show that the opposite is true); and (3) starlings' choices are not irrationally impulsive, but instead directly interpretable in terms of profitability ranking ". Certainly, the first two of these have been shown and argued in some of the authors' previous work. Thus I think that the one-sentence conclusion: "The hypothesis that preferences are built at choice time is contradicted by data on starlings' decisions by experience" has been reached previously. If not, then it needs to be explained how this goes beyond what was done before to reach this conclusion. For the third finding I would like to know more about how it differs from Shapiro et al., 2008.

I suspect that these concerns are because the authors did not have the space to properly discuss the research novelty and implications but I don't think it should be published until it is better explained. 

Reviewer #3: This paper presents an analysis of choice and response time in Starlings under different delays and magnitudes. I think these data are valuable, but the paper is confusingly written, making it difficult for me to assess its contribution. My impression is that the authors are making very strong claims about the nature of decision making beyond what the data support.

Major comments:

The exposition could use some work. A number of concepts are introduced in the first paragraph (procedural invariance, decisions from description vs experience, willingness to pay) without explanation. I think the authors should introduce these concepts more systematically.

p. 3: "A consequence of these assumptions is that response times towards a given stimulus when picked out of a choice should be shorter than when the same stimulus is encountered alone." I must confess that, based on the description of the model in the preceding paragraph, I don't see where this prediction comes from.

p. 3: what is "cross-censorship"?

p. 4: "In the pairwise comparisons where the profitability of the options was equated, the birds in fact showed the opposite: they preferred the larger-later alternative over the smaller-sooner one. This result contradicts prevalent ideas on impulsivity but supports reward rate maximization with partial account of the common time intervals." I see a few problems with this statement. First, a preference for larger-later does not mean that the animals not impulsive (they could still have a time preference expressed as a discount function), although it's not clear from this statement what exactly the alternative account is or what it predicts. Second, the reward rate maximization account is not explained enough to understand what it does or doesn't predict relative to other models. If one is going to discuss computational models in a paper, it would be very useful to actually simulate the models and fit them to data, compare the data to other models, and so on.

Showing procedural invariance using a response time measure does not necessarily contradict the results with humans where choice is the dependent measure. In any case, I think the onus is on the authors to demonstrate why you get procedural invariance in one situation/species and not another. This has to be more than just hand-waving about System 1 vs. System 2.

I found it curious that the authors contrast their work with "human experiments, which are mostly based on description rather than experience" (p. 5). But they neglect to mention that there is a now enormous human experimental literature on the description-experience gap. I see only one paper from this literature (Ludvig & Spetch, 2011) cited here. The influential review by Hertwig & Erev (2009) has already been cited over 600 times.

I think it is problematic to call response time a "rating". This is meant to draw an analogy to human desirability ratings, but it's not clear to me that these are measuring the same thing.

The finding that response time is predictive of choice is not new, at least in the human choice literature. Indeed, this is one of the key phenomena motivating sequential sampling models of choice (some of which the authors cite).

There is only one statistical analysis reported in the paper, in the caption of Figure 1, and the results are quite weak: only one comparison yields a p-value below 0.05. This does not lend strong support to the authors' claims.

Minor comments:

p. 2: "maybe endemic" -> "may be endemic"

Sometimes "procedure invariance" is used instead of "procedural invariance"

Caption of Fig 1: I find it misleading to label p<0.1 with an asterisk.

Reviewer #4: This interesting paper reports the results of a food choice experiment in Starlings. The starting point of the paper is the question whether preferences are constructed on the spot during decision making or whether they represent exist some basic approach tendencies that also guide binary choices. If the former is true, then choices should take extra time to perform compared to single-item approach decisions. However, the experiments show that this is not the case. Binary choices can be well predicted using the response times of single-item approach decisions and choices in general reflect profitability rankings of the options. The authors conclude that "irrationalities that prevail in research with humans may not show in decisions by experience protocols». 

There is a lot to like about this paper. It formally tests decision theories developed for human choice in an animal population that is not routinely tested and comes up with clean, somewhat surprising results. The methods are all fine and the results are definitively thought-provoking and should be published in some form. However, I do not fully agree with the framing and interpretation of the results and think the authors would need to change the relevant parts of the manuscript so that this is not misleading.

1) The authors claim their results show that decisions differ between experience-based protocols and protocols by description, and that human research would need to test experience-based protocols to establish whether all previous results supporting preference construction really reflect this methodical difference. The implication of this statement is that findings of human research reflect, at least in parts, linguistic decoding of option descriptions. However, this is misleading. There are numerous experience-based protocols in human research by now, for instance experiments where participants pick between depicted food items and get to actually eat one of them. I cannot see how these protocols reflect language processing any more than the disk-pecking protocol applied here. The authors would need to take this into account when comparing experiments conducted in humans and animals.

2) The authors give very little space to discussion of other possible explanations for their findings and their divergence to studies in humans. They do mention possible interspecies differences at least briefly, but they do not discuss whether their findings may be specific to pecking for food. It seems a bit of a stretch to me to draw conclusions based on the pecking protocol about e.g. human decisions where to go on holiday. The manuscript would benefit if the authors could be more precise which types of choices they talk about and which ones are probably fundamentally different and thus not affected by the present findings. 

3) The authors give very little space to the explanation of their SCM model and its predictions. In particular, the sentence that the reaction time effects result from cross-censoring of choice alternatives will be next to impossible to understand for naïve readers. The authors should at least provide references to papers explaining this model and its implications in detail when describing their predictions.

4) The authors claim that animals are very unlikely to encounter several prey at the same time. This is obviously not true for animals who hunt animals in herds and need to decide which animal to go for.

5) The authors interpret their findings in terms of the involvement of system 1 and system 2, and claim that system 2 is language based. Again, this statement is misleading. There is no direct link between system 2 and language, and it is very well possible to ponder visually-presented choice options without involving language.

---

## [Decision Letter · Decision Letter 2]

2 Jul 2020

Dear Dr Monteiro,

Thank you for submitting your revised Short Report entitled "Thinking fast and simple: how decisions are taken" for publication in PLOS Biology. I have now obtained advice from the original reviewers and have discussed their comments with the Academic Editor. 

Based on the reviews, we will probably accept this manuscript for publication, assuming that you will modify the manuscript to address the remaining points raised by reviewer 4. Regarding the title, the editors would like you to consider a more instructive one; one that conveys the central message. We suggest either “Fast decisions are not made at choice time but by ranking alternatives through direct rating” or “Starlings do not make fast decisions at choice time but rank alternatives through direct rating”, but we’d be happy to consider alternatives. We would also like you to consider moving some of the supporting figures to the main text. You have space (up to four main figures), and it would make the data more accesible. 

In addition to a clean copy of the manuscript, please also upload a 'track-changes' version of your manuscript that specifies the edits made. This should be uploaded as a "Related" file type.

Please also make sure to address the data and other policy-related requests noted at the end of this email.

We expect to receive your revised manuscript within two weeks. 

In addition to the remaining revisions and before we will be able to formally accept your manuscript and consider it "in press", we also need to ensure that your article conforms to our guidelines. A member of our team will be in touch shortly with a set of requests. As we can't proceed until these requirements are met, your swift response will help prevent delays to publication.

*Copyediting*

*Published Peer Review History*

*Early Version*

*Submitting Your Revision*

Sincerely,

Gabriel Gasque, Ph.D., 

Senior Editor

PLOS Biology

ETHICS STATEMENT:

-- Please include the full name of the IACUC/ethics committee that reviewed and approved the animal care and use protocol/permit/project license. Please also include an approval number.

DATA POLICY:

-- For figure 1d, please include all replicates AND the way in which the plotted mean and errors were derived (it should not present only the mean/average values).

-- Please also ensure that each figure legend in your manuscript include information on where the underlying data can be found (S1 Data) and ensure your supplemental data file has a legend.

Reviewer remarks:

Reviewer #1: The authors have addressed appropriately all my comments.

Reviewer #2: I think the edits and additional information provided in this revision make the points much clearer and I recommend publication of this version. 

Reviewer #3: I am satisfied by the response and revised manuscript.

Reviewer #4: The authors have been very responsive to my comments and have edited their manuscript thoroughly to address the issues I raised. I find the paper much improved now and think it will be of interest to the field (and understandable also for readers who are not experts in decision theory). Moreover, the claims are formulated much more cautiously now and are supported by the data.

I have two minor comments that the authors could address in the final revision: 

1) While it is certainly true that the SCM differs from classical DDMs in terms of its predictions, the same may not be true for the classical race models that are also being used to model value-based choices and the associated confidence in humans. It would be good if the authors could quickly mention this and briefly discuss to what degree their SCM differs from these classical race models. 

2) While the manuscript is written much more cautiously now and provides a more balanced perspective, the same cannot be said of its title. "How decisions are taken" implies that the manuscript will describe the one unifying theory of how all decisions are taken, which is clearly not the case. The authors acknowledge this themselves in their responses to my previous comments. I would therefore recommend that the authors change the title to a somewhat less grandiose version that more accurately describes the scope of their manuscript. Perhaps just "Thinking fast and simple" will do and raise the readers interest?

---

## [Editor Report · Decision Letter 3]

31 Jul 2020

Dear Dr Monteiro,

On behalf of my colleagues and the Academic Editor, Alexandre Pouget, I am pleased to inform you that we will be delighted to publish your Short Reports in PLOS Biology. 

Early Version

PRESS 

Kind regards,

Vita Usova

Publication Editor, 

PLOS Biology

on behalf of

Gabriel Gasque,

Senior Editor

PLOS Biology